

# Exploring sources of biogenic secondary organic aerosol compounds using chemical analysis and the FLEXPART model

Johan Martinsson[1,2], Guillaume Monteil[3], Moa K. Sporre[4], Anne Maria Kaldal Hansen[5], Adam Kristensson[1], Kristina Eriksson Stenström[1], Erik Swietlicki[1], Marianne Glasius[5]

[1]Division of Nuclear Physics, Lund University, Box 118, SE-22100, Lund, Sweden
[2]Centre for Environmental and Climate Research, Lund University, Ecology Building, SE-22362, Lund, Sweden
[3]Department of Physical Geography, Lund University, Lund, Box 118, SE-22100, Lund, Sweden
[4]Department of Geosciences, University of Oslo, Postboks 1022, Blindern, 0315, Oslo, Norway
[5]Department of Chemistry and iNANO, Aarhus University, Langelandsgade 140, DK-8000, Aarhus C, Denmark

*Correspondence to*: Johan Martinsson (johan.martinsson@nuclear.lu.se)

**Abstract.** Molecular tracers in secondary organic aerosols (SOA) can provide information on origin of SOA, as well as regional scale processes involved in their formation. In this study nine carboxylic acids, eleven organosulfates (OSs) and two nitrooxy organosulfates (NOSs) were determined in daily aerosol particle filter samples from Vavihill measurement station in southern Sweden during June and July 2012. Several of the observed compounds are photo-oxidation products from biogenic volatile organic compounds (BVOCs). Highest average mass concentrations were observed for carboxylic acids derived from fatty acids and monoterpenes (12.3±15.6 and 13.8±11.6 ng/m$^3$, respectively). The FLEXPART model was used to link 9 specific surface types to single measured compounds. It was found that the surface category "sea and ocean" was dominating the air mass exposure (54%) but contributed to low mass concentration of observed chemical compounds. A principal component (PC) analysis identified four components, where the one with highest explanatory power (49%) displayed clear impact of coniferous forest on measured mass concentration of a majority of the compounds. The three remaining PC's were more difficult to interpret, although azelaic, suberic, and pimelic acid were closely related to each other but not to any clear surface category. Hence, future studies should aim to deduce the biogenic sources and surface category of these compounds. This study bridges micro level chemical speciation to air mass surface exposure on the macro level.



# 1 Introduction

Carbonaceous aerosols are abundant in ambient air around the world and account for 25% of the European $PM_{10}$ mass (Fuzzi et al., 2015). The carbonaceous aerosol fraction has severe effects on human health as well as a profound effect on the Earth climate system (Dockery et al., 1993;Pope et al., 1995). During summer, carbonaceous aerosols are mainly of biogenic

origin, emitted either through primary emissions or gas-phase oxidation products from biogenic volatile organic compounds (BVOCs) (Genberg et al., 2011;Yttri et al., 2011). BVOCs are primarily emitted from plants as a tool for communication and to handle biotic and abiotic stress (Laothawornkitkul et al., 2009;Monson et al., 2013;Penuelas and Llusia, 2003;Sharkey et al., 2008). The emissions of BVOCs tend to increase with increasing temperature and photosynthetically active radiation (PAR) (Guenther et al., 1995;Guenther et al., 1993;Hakola et al., 2003). Four main compound categories dominate the global

BVOCs emissions; isoprene ($C_5H_8$), monoterpenes ($C_{10}H_{16}$), other reactive VOCs and less reactive VOCs (Laothawornkitkul et al., 2009). Isoprene is emitted from a variety of plants, however mainly from deciduous forests and shrubs which may account for more than 70% of the emissions (Guenther et al., 2006). Monoterpenes are largely emitted from coniferous trees like pine and spruce, but also from some deciduous trees, such as birch (Mentel et al., 2009). The most abundant monoterpenes in the boreal forests include α-pinene, β-pinene and limonene (Hakola et al., 2012).

Biogenic secondary organic aerosols (BSOA) are formed by photo-oxidation of BVOCs, a process which tends to lower the saturation vapor pressure of the oxidation products relative to that of the BVOCs, thus forcing the gas phase products to partition into the aerosol-phase. BSOA has been shown to dominate over combustion source aerosols during summer (Genberg et al., 2011;Yttri et al., 2011). Yttri et al. (2011) performed source apportionment at four sites in Scandinavia during August 2009 and found that the biogenic contribution to the carbonaceous aerosol dominated (69-86%) at all four

sites. Genberg et al. (2011) performed a one year source apportionment at one site in southern Sweden where they apportioned 80% of the summer-time carbonaceous aerosol to biogenic sources. Gelencser et al. (2007) also reported biogenic source dominance (63-76%) of the carbonaceous aerosol at 6 sites in south-central Europe during summer. Castro et al. (1999) observed a maximum and minimum in SOA in Europe during summer and winter, respectively. The relative SOA contribution was higher in rural forest and ocean measurement sites compared to urban sites (Castro et al., 1999).

BSOA consists of a myriad of organic compounds. Small (carbon number: C3-C6) and larger (C7-C9) dicarboxylic acids are highly hydrophilic and hygroscopic which have shown to result in potential strong climate effect due to their cloud condensation properties (Cruz and Pandis, 1998;Kerminen, 2001). Dicarboxylic acid contribution to carbon mass has been estimated to 1-3% in urban and semi-urban areas and up to 10% in remote marine areas (Kawamura and Ikushima, 1993;Kawamura and Sakaguchi, 1999). Primary aerosol sources of dicarboxylic acids in atmospheric aerosols include ocean

emissions, engine exhausts and biomass burning (Kawamura and Kaplan, 1987;Kundu et al., 2010;Mochida et al., 2003). However, the main source of dicarboxylic acids are oxidation/photo-oxidation processes of VOCs (Zhang et al., 2010). These VOC precursors may originate from both anthropogenic and biogenic sources (Mochida et al., 2003). However,





BVOCs constitute more than 50% of all atmospheric VOCs, which is approximately equal to 1150 Tg carbon y$^{-1}$ (Guenther et al., 1995;Hallquist et al., 2009).

Organosulfates (OSs) and nitrooxy organosulfates (NOSs) are low volatility SOA products that in recent years have gained increased attention due to their potential properties as tracers for atmospheric ageing of aerosols in polluted air masses (Hansen et al., 2015;Hansen et al., 2014;Kristensen, 2014;Kristensen and Glasius, 2011;Nguyen et al., 2014). Many of these compounds are formed from isoprene and monoterpene oxidation products that react with sulfuric acid in the aerosol phase (Iinuma et al., 2007;Surratt et al., 2010;Surratt et al., 2007b). Since atmospheric sulfuric acid is mainly of anthropogenic origin (Zhang et al., 2009), presence of OSs from biogenic organic precursors thus indicates an effect of anthropogenic enhancement of BSOA (Hansen et al., 2014). Recently, OSs from anthropogenic organic precursors such as alkanes and PAHs have also been discovered (Riva et al., 2016;Riva et al., 2015). Tolocka and Turpin (2012) estimated that OSs could comprise up to 10% of the total organic aerosol mass in the U.S.

Many carboxylic acids and OSs originate from biogenic sources, however, the exact vegetation types emitting the precursor are poorly explored (Mochida et al., 2003;Tolocka and Turpin, 2012). Coniferous forests, deciduous forests, arable land, pastures etc. are all examples of potential BVOCs sources. Information on specific land surface type BVOCs and BSOA emissions is potentially crucial if an increased understanding should be reached on how land-use changes will affect organic aerosol levels and composition. Yttri et al. (2011) measured one dicarboxylic acid (pinic acid), four OSs and two NOSs at four locations in Scandinavia and connected this measurement data to the FLEXPART model (Stohl et al., 2005) footprint of specific surface landscape types. They used thirteen types of surface landscapes and found that the two NOSs (MW 295 and MW 297, both formed from monoterpenes) correlated with air mass exposure to mixed forest (Yttri et al., 2011).

In this study, a comprehensive measurement campaign was conducted in order to investigate sources and levels of BSOA. 38 sequential 24h filter samples were analyzed for 9 species of carboxylic acids, 11 species of OSs and 2 species of NOSs at a rural background station in southern Sweden. FLEXPART model simulations at the time and location of the observations were then used to estimate the potential origin of the aerosols sampled.

## 2 Methods

### 2.1 Location and sampling

The Vavihill measurement station is a rural background station in southern Sweden (56°01' N, 13°09' E, 172 m.a.s.l.) within EUSAAR (European Supersites for Atmospheric Aerosol Research) and EMEP (European Monitoring and Evaluation Programme). The surrounding landscape consists of pastures, mixed forest and arable land. The largest nearby cities are Helsingborg (140 000 inhabitants), Malmö (270 000 inhabitants) and Copenhagen (1 990 000 inhabitants) at a distance of 25, 45 and 50 km, respectively. These cities are in the west and southwest direction from the measurement station. Previous observations have shown that air masses from continental Europe are usually more polluted than air masses from the north and westerly direction, i.e. Norwegian Sea and Atlantic Ocean (Kristensson et al., 2008).





38 filter samples of aerosols were collected at the Vavihill field station in southern Sweden from 10[th] of June to 18[th] of July 2012. Aerosols were collected on 150 mm quartz fibre filters (Advantec) using a high volume sampler (Digitel, DHA-80) with a $PM_1$ inlet. The filters were heated to 900°C for four hours prior to sampling, with the purpose of removing adsorbed organic compounds from the filters. The sampling air flow was 530 litres per minute and total sampling time per filter was 24 hours. Sampled filters were wrapped in aluminium foil and stored at -18°C until extraction.

## 2.2 BSOA analysis

The method for extraction and analysis is based on previous studies (Hansen et al., 2014;Kristensen and Glasius, 2011;Nguyen et al., 2014) and thus only described briefly here. For extraction each filter was placed in a beaker and spiked with 15µl of a 100 µg/ml recovery standard (camphoric acid). The filter was covered with 90% acetonitrile with 10% MilliQ water and extracted in a cooled ultrasound bath for 30 min. The extract was filtered through a Teflon filter (0.45 µm pore size, Chromafil) and evaporated until dryness using a rotary evaporator. The sample was then re-dissolved twice in 0.5 ml 3% acetonitrile, 0.1% acetic acid, and stored in a refrigerator (3-5°C) until analysis. The samples were analysed with an Ultra High Performance Liquid Chromatograph (UHPLC, Dionex) coupled to a quadrupole Time-Of-Flight Mass Spectrometer (q-TOF-MS, Bruker Daltonics) through an electro-spray ionisation (ESI) inlet. The UHPLC stationary phase was an Acquity T3 1.8µm (2.1 × 100 mm) column from Waters, and the mobile phase consisted of eluent A: 0.1% acetic acid and eluent B: Acetonitrile with 0.1% acetic acid. The operational eluent flow was 0.3 mL/min and an 18 min. multistep gradient was applied: From 1 min. to 10 min. eluent B increased from 3% to 30%, then eluent B increased to 90% during 1 min, where it was held for 1 min, before eluent B was increased further to 95% (during 0.5 min) kept here for 3.5 min. before reduction to 3% (during 0.5 min) for the remaining 0.5 min of the analysis. The ESI-q-TOF-MS was operated in negative ionisation mode with a nebulizer pressure of 3.0 bar and a dry gas flow of 8 L/min. All data were acquired and processed using Bruker Compass software. The analyzed dicarboxylic acids, OSs and NOSs are summarized in Table 1 and 2, respectively. Authentic standards were used for identification and quantification of all carboxylic acids, while OSs and NOSs were identified based on their MS/MS loss of $HSO_4^-$ (m/z = 97) and an additional neutral loss of $HNO_3$ (m/z = 63) in the case of NOSs. This work focused on identification of OSs from biogenic organic precursors, since OSs from alkanes and PAHs had not been discovered at the time of the analysis. OSs and NOSs were quantified using surrogate standards of OS 250 derived from β-pinene (synthesized in-house), octyl sulfate sodium salt (≥95% Sigma-Aldrich) or D-mannose-6-sulfate sodium salt (≥90% Sigma-Aldrich) based on their retention times in the UHPLC-q-TOF-MS system (Table 2). A linear or quadratic relationship between peak area and concentration was demonstrated for all standards and surrogates, and the correlation coefficients, $R^2$, of all calibration curves were better than 0.98 (n = 7 data points).

The analytical uncertainty was estimated to be <20% for carboxylic acids and <25% for OSs and NOSs. The uncertainty of the absolute concentrations of OSs and NOSs are higher than carboxylic acids due to lack of authentic standards.

## 2.3 Auxiliary measurements and analysis



PM$_{2.5}$ was measured with one hour time resolution using a tapered element oscillating microbalance (TEOM, Thermo, 8500 FDMS), and estimated uncertainty was less than 25%. Geographical air mass origin was analyzed with the Hybrid Single Particle Lagrangian Integrated Trajectory Model (HYSPLIT) model (Draxier and Hess, 1998;Stein et al., 2015). Gridded meteorological data from the Centre of Environmental Predictions (NCEP) Global Data Assimilation System (GDAS) were

used as input by the trajectory model. Back-trajectories were calculated at an hourly frequency 120-hour backward in time and the trajectories started 100 m above ground at the Vavihill measurement site. For each filter sample, 24 trajectories were used since the sampling time was 24 hours.

## 2.4 Source apportionment

The concentration and chemical composition of an aerosol sample depends on the trajectory of the sampled air mass in the days preceding the observation (whether or not it gets in contact with a source of aerosols or of aerosol precursors), but also on other meteorological factors such as the temperature and the amount of solar radiation (which control the chemical reactions that lead to production, destruction and transformation of aerosols), and the occurrence of precipitations, which can lead to a rapid scavenging of aerosol particles. A formal source apportionment would require a precise accounting of these

factors, which is extremely complicated and is clearly out of the scope this study. We can however already obtain valuable information just by looking at the trajectories of the air masses, and in particular, by estimating how much the air has been exposed to each land surface type.

We used the FLEXPART Lagrangian particle dispersion model in its version 10.0 (Seibert and Frank, 2004;Stohl et al., 2005) to compute seven days backward footprints for each observation. The principle is that, for each observation,

FLEXPART computes the dispersion backward in time of a large number of "particles" (i.e. small virtual air masses). A 4D (space/time) domain is defined, and the aggregated residence time of the particles in each grid box of that domain represents the sensitivity of the observation to processes within that grid box. The temporal domain is hourly (from the release of the particles to seven days before), and the spatial domain is a 0.2°x0.2° grid, ranging from 30°N to 65°N and from 2°W to 32°E. Only one surface vertical level was used, ranging from the surface to a threshold altitude level, defined as half of the

boundary layer height (computed by FLEXPART for each particle at each time step). This choice of a single surface layer means that the observations are considered insensitive to aerosol production/destruction above that layer. Accounting for such non-linear processes would require a more complex knowledge of the aerosol chemistry, and probably a more complex numerical model. The implications for this approximation are discussed in results and discussion section.

One footprint was computed for each observation, based on the dispersion seven days backward in time of 100000

FLEXPART particles. Default parameters for the FLEXPART "AERO-TRACER" tracer were used (i.e. dry and wet deposition schemes account for the removal of aerosols from the atmosphere by gravitational settling and rain washout). FLEXPART configuration files are provided in Supplementary Information.

To compute the exposure of each sample to different land surface types, we coupled the information from the footprints to the CORINE 2012 land cover map (Copernicus, 2012). CORINE 2012 is a high resolution (250x250m) map of the land





surface types in the European Union (44 land surface categories, to which we added a "sea and cean" category). The exposure $E_i$ of one observation to the land type $i$ is given by $E_i = \sum_j f_j^i R_j$, where $j$ is one pixel of the domain, $f_j^i$ is the fraction of the land surface type $i$ in that pixel, and $R_j$ is the sensitivity of the observation to that pixel (i.e. the value of the footprint at that location).

It is important to remember that since non-linear processes are not accounted for by the FLEXPART simulations, these land surface exposures are not a proper source apportionment, but are only a tool to interpret the observations.

### 2.5 Principal Component Analysis (PCA)

In order to deduce potential sources of measured BSOA compounds a principal component analysis (PCA) was performed
on measured chemical compounds together with air mass exposure to the landscape surface types derived from the FLEXPART model. The principle of PCA is that if measured parameters from the same source are strongly correlated they are treated as one principal component (PC), i.e. PCA identifies variables that have a prominent role by analysis of correlation and variance. PCA were performed by using the software SPSS (version 23, IBM).

### 3 Results and discussion
### 3.1 Variations and features in BSOA compounds

A total of 9 organic acids, 11 OSs and 2 NOSs of anthropogenic and biogenic origin were determined in the samples (Tables 1 and 2). All organic acids were quantified with authentic standards whereas the other compounds were quantified with surrogates (see experimental section). On average, the total mass of the organic chemical species from filters contributed to
0.3% ($\pm$0.2%, standard deviation) to $PM_{2.5}$. However, it is worth noticing that the particles were sampled through a $PM_1$ inlet, which most probably has excluded a considerable portion of the mass collected on filters compared to the $PM_{2.5}$ mass measured by the TEOM. On the other hand, it has been shown that $PM_1$ can comprise up to 90% of $PM_{2.5}$ in rural locations during summertime (Gomiscek et al., 2004). Since no gravimetric analysis of filters was performed, no information on the total mass loading of $PM_1$ is available.
In Table 3 and Fig. 1A concentrations of observed compounds during the sampling period are given. The compounds have been merged into groups based on their likely precursors in Fig. 1A (see Tables 1 and 2). Table 3 summarizes concentration ranges, means and standard deviations (SDs) for individual dicarboxylic acids, OSs and NOSs. In general the organic acids from monoterpenes and fatty acids dominate the total concentration over the entire period, where the concentration of acids from monoterpenes range from 1.7 to 49.0 ng m$^{-3}$ and the concentration of organic acids from fatty acids range from 0.03 to
64.1 ng m$^{-3}$. The concentration of isoprene-derived OSs ranges from 0.34 to 21.6 ng m$^{-3}$ over the sampling period and dominates over the monoterpene-derived OSs. This pattern has also been observed in other studies in the Nordic countries (Yttri et al., 2011), and is in line with high emissions of isoprene during summer. The NOSs are low in average concentration (NOS 295=0.12$\pm$0.11 ng/m$^3$, NOS 297=0.05$\pm$0.03 ng/m$^3$), and are lower than the observed mean



concentration by Yttri et al. (2011) from the summer of 2011 (NOS 295=0.74 ng/m$^3$, NOS 297=1.2 ng/m$^3$). This could be due to differences in aerosol sources and surrogate standards for quantification between the two studies.

The fatty acid derived azelaic acid was found to be the most abundant dicarboxylic acid with a concentration range from 0.03 to 55.3 ng/m$^3$ (mean=10.5±13.8 ng/m$^3$). Hyder et al. (2012) who measured 9 dicarboxylic acids in aerosol samples obtained at the Vavihill measurement station 2008-2009 also found azelaic acid to be the most prominent with peak concentration during summer (16.2 ng/m$^3$). The concentration of the anthropogenic acids is low (mean≈2 ng/m$^3$) except during the 27$^{th}$ of June and the 6$^{th}$ of July when the concentration reaches 19.6 and 16.0 ng/m$^3$, respectively. The spike in concentration of anthropogenic acids during these two days is caused by an increase in the concentration of adipic acid.

Correlations between the different compounds was investigated by Pearson correlation. All Pearson r-coefficients are given in Table 4. In general, the biogenic compounds (derived from isoprene and monoterpenes) correlated well (r≥0.8) with each other. The only exception was OS 250, which showed low to medium correlation with the other compounds.

Three dicarboxylic acids (azelaic, pimelic and suberic acid) correlated well with each other (r>0.87). It is likely that the fatty acid derived dicarboxylic acids has a different origin than isoprene and monoterpene generated acids, a conclusion that also was reached in a previous study (Hyder et al., 2012). It was expected that adipic acid would show good agreement with pimelic acid since they are both suggested to be of anthropogenic origin. However, this correlation was poor (r=0.16) and is believed to be explained by two strong concentration peaks in adipic acid (27$^{th}$ of June and 6$^{th}$ of July, Fig. 1A) with no corresponding peak in pimelic acid.

**3.2 Air mass surface exposure**

Figure 1B displays the exposures of the samples to the nine largest surface categories as percentage contribution and Tables 5 and 6 present the mean exposures and a correlation matrix for the investigated surface types. These surface categories are explained in more detail in the supplemental information. The "sea and ocean" category is dominating the exposure with an average of 56% (±16%). This is hardly surprising since a majority of the incoming air mass is from the westerly region where the North Atlantic Ocean, North Sea and Norwegian Sea are situated. The second most common surface exposure is from "non-irrigated arable land" (mean=19% ±8%). This is a common land type in continental Europe which is anti-correlated (r=-0.84) to the "sea and ocean" surface category. The fact that several land-based surface categories anti-correlated to the "sea and ocean" category may be an indicator of the model working properly. The category "other" has a significant contribution to the total exposure (mean=8% ±3%), but it groups 34 surface categories and is therefore difficult to interpret beyond the common fact that all these categories are land masses. It is important to remember that these exposures should not be read as a representation of the contribution of the land surface types to the production of the aerosols measured. For that, an estimation of the aerosol production (or transformation) associated to each surface category would be required. However, correlating the land surface exposures to the measured aerosol time series can provide an indication on the origin of the aerosols.



During a period of increased concentrations of molecular BSOA compounds (6[th] to 8[th] of July) the air mass was more exposed to land surface categories such as "non-irrigated arable land", "coniferous forest", "broad leaved forest" and "pastures" on the expense of "sea and ocean" (Fig. 1A-B). Further, the category "other" is also increased during this particular period. Within the "other" category, "mixed forest", "complex cultivation patterns", "land principally occupied by

5 agriculture, with significant areas of natural vegetation" and "transitional woodland/shrub" are dominating (more information about the surface categories can be found on the CORINE database homepage) (EEA, 2016). This particular concentration increase is caused by the fatty acid-derived organic acids, monoterpene-derived organic acids and isoprene-derived OSs (Fig. 1A). The concentration of $PM_{2.5}$ does not provide any explanation of the cause of the high concentrations, since $PM_{2.5}$ is in general high during the entire campaign period. Both the HYSPLIT and FLEXPART model revealed that

arriving air masses during this period mainly had an origin from continental Europe (Fig. 2). As stated earlier, it has been observed that air masses arriving from this direction usually carry more PM and OSs than from other directions (Nguyen et al., 2014;Kristensson et al., 2008).

The period of increased concentrations of molecular BSOA compounds (6[th] to 8[th] of July) are in large contrast to the "clean periods" observed during 12[th]-16[th] of June and 16[th]-18[th] of July (Fig. 1A-B). In particular, the latter period shows very low

values of molecular BSOA compounds and a corresponding "sea and ocean" exposure of 79-86%. Hence, "sea and ocean" exposure does not seem to contribute to the measured mass of molecular BSOA compounds. Similarly, the "non-irrigated arable land" contributes to a significant fraction during 16[th]-18[th] of July (8-12%) and most probably does not contribute to the mass of measured BSOA species either.

**3.3 Connection between surface type and measured species**

To further investigate the impact of surface types on measured BSOA species a principal component analysis (PCA) was conducted as described in the method section. To our knowledge, the measured SOA species are derived from four possible precursor sources: anthropogenic, fatty acids, isoprene and monoterpene (Tables 1 and 2), hence a 4 PC VARIMAX-rotated solution was chosen. This solution explained 80.3% of the total variance. Table 7 shows the individual parameter

contribution to the respective PC. PC1 accounts for 49.1% of the total variance and has strong positive contributions from several of the monoterpene derived dicarboxylic acids and both monoterpene and isoprene derived OSs and NOSs. The strongest positive surface category in PC1 is "coniferous forest", suggesting that the species with a bold number in PC1 within Table 7 are originating, or that their mass concentration have a positive response, from coniferous forest. Coniferous forests are mainly known as large-scale emitters of monoterpenes. Despite this, the PCA illustrates that isoprene oxidation

products are positively correlated to this surface category. Steinbrecher et al. (1999) observed negligible emissions of isoprene from common conifers as Scots pine (*Pinus sylvestris*) and common juniper (*Juniperus communis*). However, they found significant emissions from Norway spruce (*Picea abies*) which may explain some of the isoprene derived compounds in this study. Although the less strong positive contribution of 0.53, isoprene emitting "broad leaved forest" may also have contributed to the above described pattern in PC1.





PC2 accounts for 14.9% of the total variation and can roughly be classified as surface categories with low contribution to measured BSOA compounds. Six of the ten investigated surface categories show strong positive contribution to PC2 while many of the measured compounds show low and in some cases negative contribution to PC2. The observed pattern of high "sea and ocean" and "non-irrigated arable land" exposure when the mass concentration of BSOA compounds was low,

further strengthens the explanation of PC2.

PC3 accounts for 9.3% of the total variance. The main contributors are suberic acid, azelaic acid and pimelic acid. They are all similar in chemical structure, although suberic and azelaic acid probably originate from fatty acids while pimelic acid likely is of anthropogenic origin (Table 1). Further, azelaic acid has been found to be involved in the triggering of the plant immune system (Jung et al., 2009). Hyder et al. (2012), who also found these three acids to be highly correlated in ambient

aerosol, inferred that pimelic acid was either produced from the same source as suberic and azelaic acid or that pimelic acid is produced by continued oxidation of suberic and azelaic acid down to lower carbon numbered acids. None of the land surface categories displayed high contribution to PC3: "broad leaved forest" had the highest contribution of 0.21 while the other forest category, "conifer forest", had a one order of magnitude lower contribution of -0.04. Hence, it is possible that broad leaved forests are more important for higher carbon (C7-C9) carboxylic acid production than coniferous forests.

PC4 accounted for 6.9% of the total variance and is harder to interpret than the previous three PC's. The anthropogenic derived adipic acid has a positive PC contribution (0.59) as well as the surface categories "sparsely vegetated areas" (0.86) and "moors and heath" (0.85). The used land cover maps reveals that both "sparsely vegetated areas" and "moors and heath" are mainly found in Norway and northern Sweden, i.e. in the north and north-westerly direction of Vavihill measurement station. The overall interpretation of PC4 is difficult since adipic acid are thought to be of anthropogenic origin but, in this

case, seem to correlate with landscape surface types that are sparsely populated and are associated with low human activity (i.e. "sparsely vegetated areas" and "moors and heath"). It is questionable whether the pre-assumed anthropogenic acids, adipic and pimelic acid, actually share any kind of origin since neither this study nor the study by Hyder et al. (2012) found any strong correlation between these two dicarboxylic acids. Nevertheless, future studies should repeat the presented methodology to focus on heavy anthropogenic influenced surface categories (i.e. cities, industries etc.) and their impact on

anthropogenic acids and newly discovered anthropogenic OSs (Riva et al., 2016;Riva et al., 2015).

### 3.4 Uncertainties and limits

In this study, our analysis approach relies on two steps: first the calculation of the exposures, using FLEXPART, and then the estimation of land type contributions using a PCA analysis. Both steps suffer from uncertainties which limit the

robustness of our results:

The longer the back-trajectories used in FLEXPART, the larger the error is likely to be. On the other hand, shorter back-trajectories lead to neglecting a larger proportion of "older" aerosols. We tested the impact of the footprint length choice on the exposure time series by repeating the analysis with footprints of 3 and 5 days (instead of 7 days in our default setup).




Overall, the exposures are not significantly affected, except for the exposure to the "sea and ocean" surface type during the 8-10 July peak, which show an uncertainty of 6% (Fig. S1).

The calculation of the observation exposures is based on the assumption that the measured aerosol compositions scale linearly with the aerosol production within the back-plume of the observation. This is not the case in reality: processes such as coagulation, nucleation, chemical reactions between aerosols and surrounding reactive gas species, photo-dissociation and wet and dry deposition (removal of aerosols from the atmosphere by the rain and by gravitational settling) alter the aerosol composition and concentration all along the air mass trajectory. Our approach also ignores the influence of aerosol particles (or precursors) older than seven days on the observations. Accounting adequately for all these processes would require a comprehensive (much heavier) aerosol model, which is totally out of the scope of this study. This mainly means that our approach cannot be used to quantify the aerosol production associated to, for example, a specific forest type. It nonetheless provides valuable qualitative information that could probably not be obtained with simpler single air-mass trajectory analysis such as the ones computed with the HYSPLIT model (Fig. 2).

The main limit to the PCA analysis is the shortness of the time series. In particular, there is only one strong event during the campaign (6-8 July), which is not enough for drawing strong conclusions. Our study can however be regarded as a proof-of-concept: computing FLEXPART footprints is relatively easy and lightweight, and could be performed routinely. The conclusions of a PCA analysis are likely to be a lot more robust with longer time series, and/or multi-sites observation campaigns (provided that the footprints of the different sites overlap sufficiently).

## 4 Conclusions

Nine carboxylic acids along with eleven organosulfates (OSs) and two nitrooxy organosulfates (NOSs) were analyzed from 38 daily aerosol samples sampled at Vavihill measurement station in southern Sweden during June and July 2012. Most of the measured compounds can be considered as photo-oxidation products from biogenic volatile organic compounds (BVOCs), hence derived from terrestrial plants. The FLEXPART model was used to identify exposure of the aerosol samples to several different surface categories. For easier interpretation, the study was focused on four potential source-specific components using 22 chemical species and the nine largest surface categories. The "sea and ocean" category was found to dominate the exposure, and other important categories were "non-irrigated arable land" and "pastures". A principal component analysis (PCA) of four principal components (PC) was used to explore the impact and connection of surface categories on mass concentration of measured biogenic secondary organic aerosol compounds. It was found that coniferous forest had a positive effect on several of the measured monoterpene-derived compounds. The remaining three PCs were harder to interpret, however future studies should aim to investigate the sources of azelaic, suberic and pimelic acids which dominate in mass concentration but showed no clear correlation to surface categories.

This study demonstrates the interest of using an atmospheric transport model in aerosol source apportionment on specific chemical compounds. With the presented methodology it is possible to connect single chemical tracer compounds to potential local and long range aerosol sources, i.e. surface categories. Further, this FLEXPART application enables detailed





investigations on how natural and anthropogenic land-use changes may affect the mass concentration and chemical composition of ambient aerosol.

**5 Data availability**

All data are accessible through the supporting information.

**Author contribution**

Johan Martinsson designed the study, compiled all data, performed the PCA and wrote most of the paper. Guillaume Monteil ran the FLEXPART simulations. Moa K. Sporre ran the HYSPLIT simulations. Anne Maria Kaldal Hansen and Marianne

Glasius ran the chemical analysis. Adam Kristensson, Erik Swietlicki and Kristina Eriksson Stenström assisted in the writing process.

**Acknowledgements**

This work was supported by the Swedish Research Council FORMAS (project 2011-743).

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





**Table 1: Analyzed organic acids in the Vavihill aerosol samples. Measured m/z, molecular formula, possible molecular structure, suggested precursor and assigned precursor class. a) Hatakeyama et al. (1987), b) Stephanou and Stratigakis (1993), c) Kawamura and Gagosian (1987), d) Szmigielski et al. (2007), e) Ma et al. (2007), f) Claeys et al. (2009).**

| Precursor class | Name | Measured m/z | Molecular formula | Possible structure | Suggested precursor |
|---|---|---|---|---|---|
| **Anthropogenic** | Adipic acid | 145.050 | $C_6H_{10}O_4$ | | Cyclohexene[a] |
| | Pimelic acid | 159.065 | $C_7H_{12}O_4$ | | Cycloheptene[a] |
| **Fatty acid-derived** | Suberic acid | 173.081 | $C_8H_{14}O_4$ | | Unsaturated fatty acid[b,c] |
| | Azelaic acid | 187.097 | $C_9H_{16}O_4$ | | Unsaturated fatty acid[b,c] |
| **1. generation Monoterpene** | Pinic acid | 185.081 | $C_9H_{14}O_4$ | | α-/β-pinene[d,e] |
| | Pinonic acid | 183.102 | $C_{10}H_{16}O_3$ | | α-/β-pinene[d,e] |
| | Terpenylic acid | 171.065 | $C_8H_{12}O_4$ | | α-pinene[f] |
| **2. generation Monoterpene** | 3-methyl-1,2,3-butane-tricarboxylic acid (MBTCA) | 203.055 | $C_8H_{12}O_6$ | | α-pinene[d] |
| | Diaterpenylic acid acetate (DTAA) | 231.086 | $C_{10}H_{16}O_6$ | | α-pinene[f] |



**Table 2: Analyzed organosulfates (OSs) and nitrooxy organosulfates (NOSs) in the Vavihill aerosol samples. Measured m/z, molecular formula, possible molecular structure, suggested precursor and assigned precursor class. a) Surratt et al. (2007a), b) Schindelka et al. (2013), c) Olson et al. (2011), d) Shalamzari et al. (2013), e) Gomez-Gonzalez et al. (2008), f) Surrat et al. (Surratt et al., 2008), g) Hettiyadura et al. (2015), h) Surratt et al. (Surratt et al., 2010). The OSs and NOSs were quantified with D-mannose 6-sulfate (1), β-pinene OS 250 (2) or octyl sulfate (3).**

| Precursor class | Name | Measured m/z | Molecular formula | Possible structure | Suggested precursor |
|---|---|---|---|---|---|
| Isoprene/ Anthropogenic | OS 140[1] | 138.970 | $C_2H_4O_5S$ | | Glycolaldehyde[a] |
| | OS 154[1] | 152.985 | $C_3H_6O_5S$ | | Hydroxyacetone[a]/Methacrolein[b]/ Methyl vinyl ketone[b] |
| | OS 156[1] | 154.961 | $C_2H_4O_6S$ | | Glycolic acid[c,d]/ Methyl vinyl ketone[b] |
| | OS 170[1] | 168.979 | $C_3H_6O_6S$ | | Methylglycolic acid[c,d] |
| | OS 200[1] | 198.991 | $C_4H_8O_7S$ | | 2-methylglyceric acid[a,e] |
| Isoprene | OS 212[1] | 210.991 | $C_5H_8O_7S$ | | Isoprene[f,g] |
| | OS 214[1] | 213.007 | $C_5H_{10}O_7S$ | More isomers | Isoprene[f] |
| | OS 216[1] | 215.021 | $C_5H_{12}O_7S$ | | $C_5$-epoxydiols from isoprene (IEPOX)[h] |
| Monoterpene | OS 250[2] | 249.080 | $C_{10}H_{18}O_5S$ | More isomers | α-/β-pinene and limonene[f] |
| | OS 268[2] | 267.053 | $C_9H_{16}O_7S$ | | Limonene[f] |
| | OS 280[2] | 279.054 | $C_{10}H_{16}O_7S$ | | α-/β-pinene[f] |
| Monoterpene NOS | NOS 295[3] | 294.062 | $C_{10}H_{17}O_7NS$ | More isomers | α-/β-pinene, Limonene[a,f] |
| | NOS 297[2] | 296.044 | $C_9H_{15}O_8NS$ | More isomers | Limonene[f] |





**Table 3: Ranges of concentrations, means and standard deviation (SD) of the analyzed compounds in aerosol samples collected at the Vavihill measurement station 10<sup>th</sup> June to 18<sup>th</sup> of July 2012.**

| Compound | N | Minimum (ng/m$^3$) | Maximum (ng/m$^3$) | Mean (ng/m$^3$) | ±SD (ng/m$^3$) |
|---|---|---|---|---|---|
| Adipic acid | 36 | 0.03 | 19.27 | 1.76 | 3.87 |
| Pimelic acid | 36 | 0.02 | 1.21 | 0.38 | 0.28 |
| Suberic acid | 31 | 0.05 | 9.03 | 2.45 | 2.42 |
| Azelaic acid | 35 | 0.03 | 55.27 | 10.52 | 13.83 |
| Pinic acid | 38 | 0.28 | 4.71 | 1.31 | 1.04 |
| Pinonic acid | 38 | 0.82 | 10.66 | 2.89 | 2.00 |
| Terpenylic acid | 38 | 0.72 | 8.86 | 2.57 | 1.87 |
| DTAA | 38 | 0.04 | 5.67 | 0.84 | 1.23 |
| MBTCA | 38 | 0.38 | 29.42 | 6.18 | 7.00 |
| OS 140 | 38 | 0.02 | 0.28 | 0.11 | 0.07 |
| OS 154 | 38 | 0.15 | 2.95 | 0.76 | 0.64 |
| OS 156 | 32 | 0.02 | 2.35 | 0.65 | 0.61 |
| OS 170 | 38 | 0.08 | 0.78 | 0.33 | 0.17 |
| OS 200 | 38 | 0.06 | 2.02 | 0.41 | 0.40 |
| OS 212 | 38 | 0.16 | 4.63 | 0.91 | 0.95 |
| OS 214 | 38 | 0.06 | 3.08 | 0.50 | 0.58 |
| OS 216 | 38 | 0.06 | 5.83 | 0.63 | 1.07 |
| OS 250 | 38 | 0.02 | 3.48 | 0.51 | 0.64 |
| OS 268 | 38 | 0.01 | 0.48 | 0.13 | 0.12 |
| OS 280 | 32 | 0.01 | 0.70 | 0.09 | 0.17 |
| NOS 295 | 38 | 0.02 | 0.53 | 0.12 | 0.11 |
| NOS 297 | 37 | 0.01 | 0.18 | 0.05 | 0.03 |





**Table 4: Correlation matrix displaying the Pearson product-moment coefficient (r) for measured chemical species. Colours represent degree of correlation: yellow: |0.7-0.8|; green: |0.8-0.9|; red: |0.9-1.0|.**

| | Adipic acid | Pimelic acid | Suberic acid | Azelaic acid | Pinic acid | Pinonic acid | Terpenylic acid | DTAA | MBTCA | OS 140 | OS 154 | OS 156 | OS 170 | OS 200 | OS 212 | OS 214 | OS 216 | OS 250 | OS 268 | OS 280 | NOS 295 | NOS 297 |
|---|---|---|---|---|---|---|---|---|---|---|---|---|---|---|---|---|---|---|---|---|---|---|
| Adipic acid | | | | | | | | | | | | | | | | | | | | | | |
| Pimelic acid | 0.16 | | | | | | | | | | | | | | | | | | | | | |
| Suberic acid | 0.02 | 0.95 | | | | | | | | | | | | | | | | | | | | |
| Azelaic acid | 0.01 | 0.87 | 0.95 | | | | | | | | | | | | | | | | | | | |
| Pinic acid | 0.25 | 0.20 | 0.01 | 0.20 | | | | | | | | | | | | | | | | | | |
| Pinonic acid | 0.05 | 0.02 | 0.32 | 0.00 | 0.81 | | | | | | | | | | | | | | | | | |
| Terpenylic acid | 0.33 | 0.35 | 0.18 | 0.40 | 0.80 | 0.39 | | | | | | | | | | | | | | | | |
| DTAA | 0.35 | 0.29 | 0.18 | 0.37 | 0.66 | 0.20 | 0.89 | | | | | | | | | | | | | | | |
| MBTCA | 0.32 | 0.22 | 0.06 | 0.26 | 0.71 | 0.29 | 0.94 | 0.92 | | | | | | | | | | | | | | |
| OS 140 | 0.13 | 0.41 | 0.27 | 0.50 | 0.47 | 0.06 | 0.90 | 0.83 | 0.70 | | | | | | | | | | | | | |
| OS 154 | 0.33 | 0.36 | 0.22 | 0.43 | 0.67 | 0.19 | 0.92 | 0.94 | 0.93 | 0.82 | | | | | | | | | | | | |
| OS 156 | 0.22 | 0.36 | 0.26 | 0.34 | 0.62 | 0.21 | 0.83 | 0.87 | 0.84 | 0.76 | 0.92 | | | | | | | | | | | |
| OS 170 | 0.24 | 0.24 | 0.00 | 0.31 | 0.58 | 0.21 | 0.77 | 0.73 | 0.80 | 0.84 | 0.86 | 0.83 | | | | | | | | | | |
| OS 200 | 0.27 | 0.32 | 0.19 | 0.41 | 0.58 | 0.10 | 0.81 | 0.93 | 0.86 | 0.80 | 0.96 | 0.93 | 0.84 | | | | | | | | | |
| OS 212 | 0.34 | 0.35 | 0.22 | 0.43 | 0.65 | 0.17 | 0.86 | 0.97 | 0.88 | 0.76 | 0.97 | 0.92 | 0.81 | 0.98 | | | | | | | | |
| OS 214 | 0.33 | 0.30 | 0.18 | 0.38 | 0.61 | 0.15 | 0.80 | 0.96 | 0.82 | 0.70 | 0.92 | 0.89 | 0.74 | 0.97 | 0.98 | | | | | | | |
| OS 216 | 0.33 | 0.26 | 0.21 | 0.33 | 0.50 | 0.06 | 0.65 | 0.89 | 0.68 | 0.55 | 0.80 | 0.79 | 0.57 | 0.89 | 0.91 | 0.96 | | | | | | |
| OS 250 | 0.20 | 0.00 | 0.00 | 0.00 | 0.48 | 0.26 | 0.56 | 0.45 | 0.51 | 0.51 | 0.54 | 0.59 | 0.63 | 0.53 | 0.50 | 0.45 | 0.31 | | | | | |
| OS 268 | 0.19 | 0.12 | 0.00 | 0.08 | 0.63 | 0.36 | 0.80 | 0.72 | 0.87 | 0.63 | 0.78 | 0.67 | 0.72 | 0.71 | 0.69 | 0.64 | 0.45 | 0.55 | | | | |
| OS 280 | 0.38 | 0.24 | 0.15 | 0.14 | 0.56 | 0.08 | 0.84 | 0.83 | 0.93 | 0.65 | 0.88 | 0.77 | 0.76 | 0.78 | 0.82 | 0.75 | 0.66 | 0.39 | 0.75 | | | |
| NOS 295 | 0.00 | 0.00 | 0.00 | 0.00 | 0.56 | 0.56 | 0.53 | 0.38 | 0.62 | 0.35 | 0.44 | 0.44 | 0.50 | 0.32 | 0.33 | 0.27 | 0.09 | 0.33 | 0.70 | 0.55 | | |
| NOS 297 | 0.01 | 0.14 | 0.00 | 0.14 | 0.53 | 0.35 | 0.67 | 0.57 | 0.77 | 0.57 | 0.68 | 0.61 | 0.68 | 0.59 | 0.57 | 0.50 | 0.31 | 0.42 | 0.85 | 0.70 | 0.88 | |



**Table 5: Ranges, means and standard deviations (SD) of the FLEXPART surface type exposure of incoming air masses during 10[th] June to 18[th] of July 2012.**

| Surface type | N | Minimum (%) | Maximum (%) | Mean (%) | ±SD (%) |
|---|---|---|---|---|---|
| Pasture | 38 | 0 | 13 | 4.4 | 3.6 |
| Discontinous urban fabric | 38 | 1 | 7 | 2.6 | 1.7 |
| Non-irrigated arable land | 38 | 7 | 35 | 18.8 | 8.3 |
| Sparsely vegatated areas | 38 | 0 | 3 | 0.4 | 0.9 |
| Broad leaved forest | 38 | 0 | 8 | 2.6 | 1.7 |
| Lakes and ponds | 38 | 0 | 3 | 0.9 | 0.6 |
| Moors and heath | 38 | 0 | 3 | 0.5 | 0.7 |
| Coniferous forest | 38 | 0 | 22 | 5.5 | 5.2 |
| Sea and ocean | 38 | 24.6 | 86 | 56.0 | 16.3 |
| Other | 38 | 3 | 15 | 8.3 | 3.2 |





**Table 6: Correlation matrix displaying the Pearson product-moment coefficient (r) for surface types. Colours represent degree of correlation: yellow: |0.7-0.8|; green: |0.8-0.9|; red: |0.9-1.0|.**

| | Pasture | Discontinous urban fabric | Non-irrigated arable land | Sparsely vegatated areas | Broad leaved forest | Lakes and ponds | Moors and heath | Coniferous forest | Sea and ocean | Other |
|---|---|---|---|---|---|---|---|---|---|---|
| Pasture | | | | | | | | | | |
| Discontinous urban fabric | 0.92 | | | | | | | | | |
| Non-irrigated arable land | 0.89 | 0.9 | | | | | | | | |
| Sparsely vegatated areas | -0.47 | -0.42 | -0.49 | | | | | | | |
| Broad leaved forest | 0.48 | 0.32 | 0.53 | -0.13 | | | | | | |
| Lakes and ponds | 0 | -0.12 | -0.13 | 0.18 | 0.2 | | | | | |
| Moors and heath | -0.46 | -0.4 | -0.47 | 0.98 | -0.17 | 0.14 | | | | |
| Coniferous forest | -0.17 | -0.31 | -0.22 | 0.23 | 0.43 | 0.8 | 0.17 | | | |
| Sea and ocean | -0.84 | -0.78 | -0.84 | 0.27 | -0.73 | -0.31 | 0.28 | -0.28 | | |
| Other | 0.59 | 0.57 | 0.53 | -0.16 | 0.42 | 0.29 | -0.18 | 0.23 | -0.77 | |





**Table 7: Principal component (PC) loadings. The loadings display the variation (between -1 and 1) explained by the PC. Numbers in bold indicates absolute number >0.6. PC1 explained 49.1%, PC2 14.9%, PC3 9.3% and PC4 6.9%.**

| | Principal Component | | | |
| | 1 | 2 | 3 | 4 |
|---|---|---|---|---|
| **Adipic acid** | 0.37 | -0.25 | 0.08 | 0.59 |
| **Pimelic acid** | 0.24 | 0.20 | **0.75** | -0.21 |
| **Suberic acid** | 0.20 | 0.26 | **0.82** | -0.19 |
| **Azelaic acid** | 0.21 | 0.39 | **0.74** | -0.17 |
| **Pinic acid** | **0.70** | -0.04 | -0.25 | 0.14 |
| **Pinonic acid** | 0.19 | -0.15 | -0.37 | 0.16 |
| **Terpenylic acid** | **0.88** | 0.29 | -0.11 | 0.04 |
| **DTAA** | **0.93** | 0.24 | 0.04 | 0.09 |
| **MBTCA** | **0.89** | 0.28 | -0.26 | -0.02 |
| **OS 140** | **0.76** | 0.30 | 0.12 | -0.41 |
| **OS 154** | **0.96** | 0.22 | 0.04 | -0.10 |
| **OS 156** | **0.93** | 0.06 | 0.04 | -0.14 |
| **OS 170** | **0.79** | 0.20 | -0.17 | -0.28 |
| **OS 200** | **0.92** | 0.18 | 0.10 | -0.12 |
| **OS 212** | **0.95** | 0.18 | 0.13 | -0.01 |
| **OS 214** | **0.92** | 0.13 | 0.15 | 0.04 |
| **OS 216** | **0.87** | 0.03 | 0.26 | 0.11 |
| **OS 250** | 0.48 | -0.06 | -0.38 | -0.06 |
| **OS 268** | **0.67** | 0.24 | -0.51 | -0.18 |
| **OS 280** | **0.87** | 0.13 | -0.20 | -0.05 |
| **NOS 295** | 0.43 | 0.16 | **-0.69** | -0.25 |
| **NOS 297** | 0.59 | 0.28 | -0.48 | -0.35 |
| **Pastures** | 0.22 | **0.85** | 0.15 | -0.37 |
| **Discontinous urban fabric** | -0.02 | **0.92** | 0.12 | -0.29 |
| **Non-irrigated arable land** | 0.20 | **0.94** | 0.10 | -0.14 |
| **Broad leaved forest** | 0.53 | **0.77** | 0.21 | 0.11 |
| **Sparsely vegetated areas** | -0.11 | -0.10 | -0.18 | **0.86** |
| **Lakes and ponds** | **0.76** | 0.34 | 0.02 | 0.42 |
| **Moors and heath** | -0.16 | -0.04 | -0.23 | **0.85** |
| **Coniferous forest** | **0.79** | 0.35 | -0.04 | 0.39 |
| **Sea and ocean** | 0.37 | **0.62** | 0.27 | 0.34 |
| **Other** | **0.60** | **0.65** | 0.19 | 0.19 |





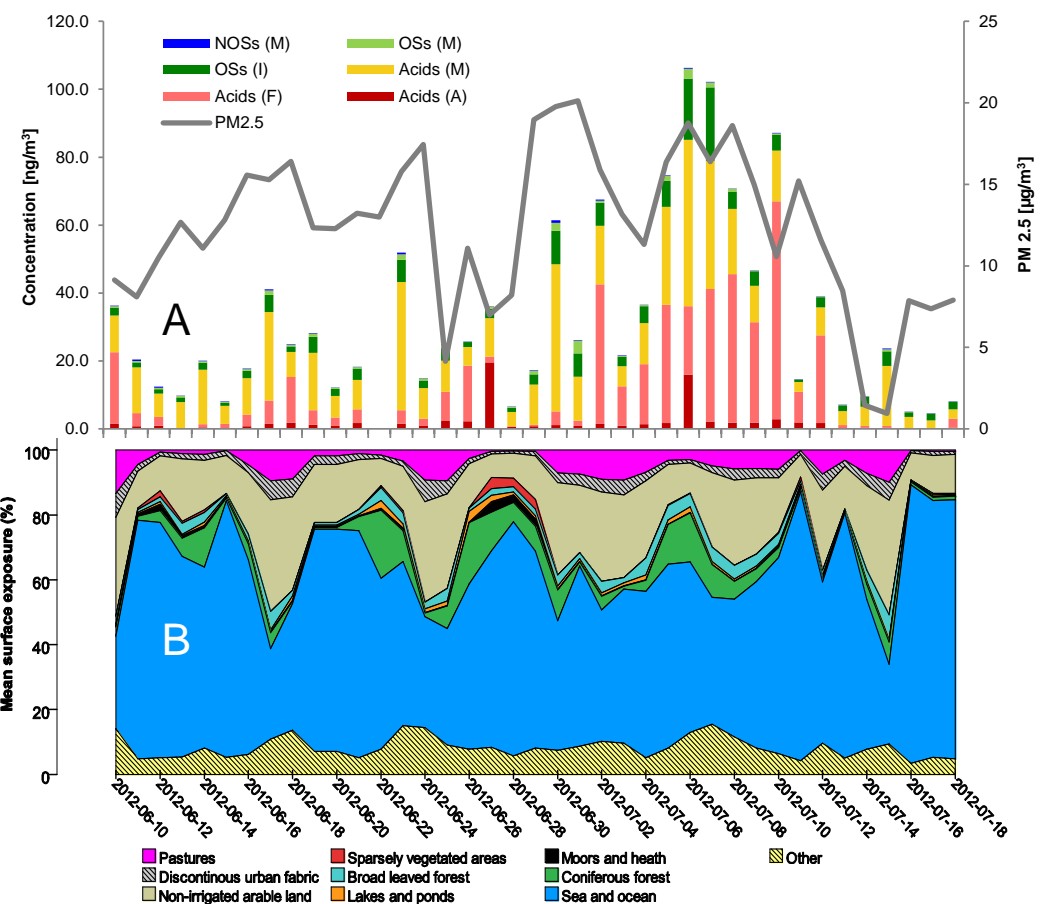

**Figure 1: A)** Total concentration of all measured carboxylic acids, organosulfates (OSs) and nitrooxy organosulfates (NOSs) in PM$_1$ collected at the Vavihill measurement station. The thick grey line displays the PM$_{2.5}$ concentration. Capital letters in parenthesis in the legend is the precursor class given in Table 1 and 2. A=Anthropogenic, F=Fatty acid, I=Isoprene and M=Monoterpenes. **B)** FLEXPART generated mean exposure from the nine mean largest surface categories. The exposure is a mean of 3, 5 and 7 days back trajectories. The category "Other" represents the remaining 34 surface categories. More detailed information on the surface categories can be found in the supplemental information.



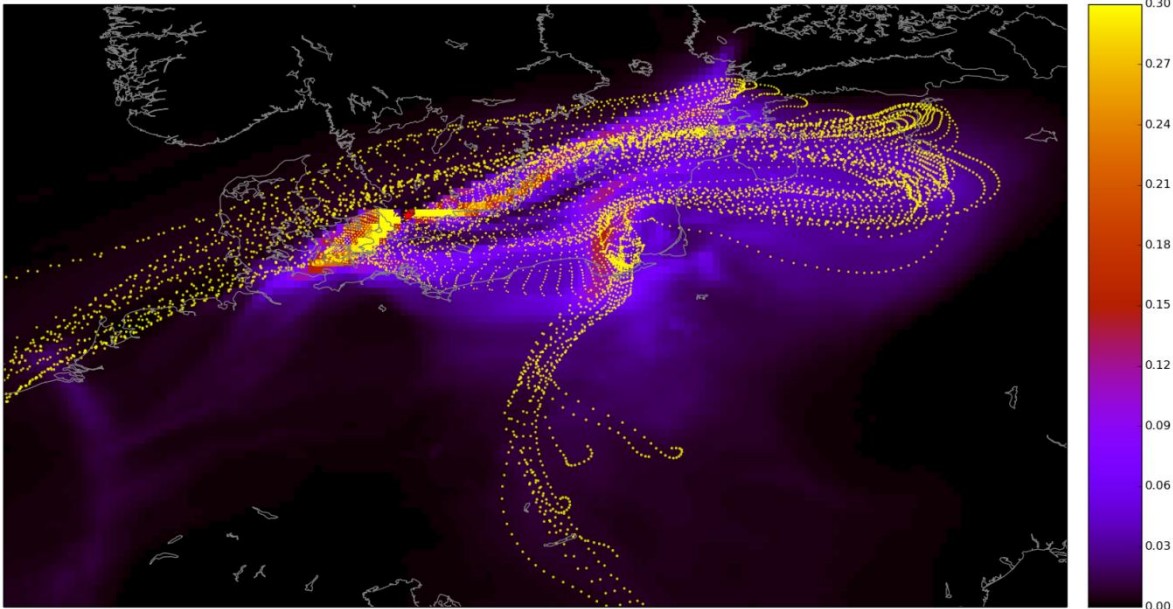

**Figure 2: 120 hour back-trajectory air mass covering the concentration peak dates; 6-8<sup>th</sup> July. FLEXPART are shown in shaded colors while HYSPLIT are displayed by yellow dotted trajectories. The dots size increase with the air mass age. The colorbar displays the FLEXPART footprint, normalized to 1 (the color range has been limited to 0-0.3 to highlight grid points with low but a non-zero contribution). Together, the grid points with a value larger than 0.1 contribute 17% of the total sensitivity while grid boxes with a value larger than 0.01 contribute 81% of the total sensitivity. 120 h back-trajectory was chosen for easier interpretation of the illustration.**