# Peer review of "Exploring sources of biogenic secondary organic aerosol compounds using chemical analysis and the FLEXPART model"

_Atmospheric Chemistry and Physics, 2017_

## Referee Comment (RC1) · Anonymous Referee #1 · 27 Feb 2017

**Review: Atmos. Chem. Phys. Discuss. ; acp-2017-90 (Martinsson et al.)**

**General comments:**

This manuscript explores the connection between chemical speciation at the micro level and air mass surface exposure at the macro level. The study is based on a comprehensive chemical data set of organic species. An expected outcome of the study is that that the surface category "coniferous forest" had a clear impact on the mass concentration of the measured compounds, whereas the surface category "sea and ocean" only had a low explanatory power. As the authors state, the biogenic source and surface origin of the dicarboxylic acids, azelaic acid, suberic acid and pimelic acid, which are closely related, is not clear, and should be the focus of future studies.

**Specific comments:**

Page 2 – line 14: Another monoterpene to be considered is $\Delta^3$-carene (Räisänen et al., 2013).

Page 2 – 25: The number of carbon atoms in a molecule should be indicated with a subscript: $C_3$-$C_6$, $C_7$-$C_9$.

Page 4 – line 23: A correction is needed here: "….. their MS/MS formation of $HSO_4^-$ (*m/z* 97) ……….. (63 u)….". Note that according to the IUPAC guidelines for terms related to mass spectrometry "*m/z* " should be in italic font (Murphy et al. 2013). Furthermore, the neutral loss of $HNO_3$ corresponds to 63 "mass units", abbreviated as "u".

Page 16 – Table 1: The structure of MBTCA should be corrected as follows:

**Technical corrections:**

Page 2 – line 6 and many places elsewhere: a space is missing after ";" in the cited references.

Page 2 – line 16: …. gas-phase products …..

Page 2 – line 17: ….. in the aerosol phase.

Page 3 – line 3: …. low-volatility …..

Page 3 – line 20: ….. a one-year study ….

Page 4 – line 2: …. high-volume …..

Page 4 – lines 3, 5 and 12: a space is missing before "$^oC$".

Page 4 – line 9: …. 15 µL ……

Page 4 – lines 16, 17 and 18: ….. min ….. ("minutes" is abbreviated as "min").

Page 4 – lines 9 and 11: ….. mL …….

Page 4 – line 19: The abbreviation "MS" stands for the technique "mass spectrometry" and not for the instrument. Thus: "The ESI-q-TOF-MS instrument …." (see Murphy et al., 2013).

Page 5 – line 13: ….. of precipitation, …..

Page 7 – line 3: …. fatty acid-derived …..

Page 7 – line 12/13: …. fatty acid-derived …..

Page7 – line 13: …. have a different origin than isoprene- and monoterpene-generated acids, …..

Page 8 – line 3: …. "broad-leaved forest"…..

Page 8 – line 23: …… and monoterpenes …..

Page 8 – line 26: ….. monoterpene-derived …….. both monoterpene- and isoprene-derived …..

Page 8 – line 33: …. "broad-leaved forest"…..

Page 9 – line 12: …. "broad-leaved forest"…..

Page 9 – line 14: ….. ($C_7$-$C_9$) …. (see specific comment above).

Page 9 – line 19: …… is thought to …..

Pages 11 – 14: Titles of journal articles should not be capitalized.

Table 2 – legend: . Measured *m/z*, ……. f) Surratt et al. (2008), ………… h) Surratt et al. (2010).

References:

K. K. Murray,  R. K. Boyd, M. N. Eberlin, G. J. Langley, L. Li, Y. Naito. Definitions of terms relating to mass spectrometry (IUPAC Recommendations 2013. Pure Appl. Chem., 85, 1515-1609, 2013.

T. Räisänen, A. Ryyppö, S. Kellomäki. Effects of elevated $CO_2$ and temperature on monoterpene emission of Scots pine (Pinus sylvestris L.). Atmos. Environ. 2008, 42, 4160.

---

## Short Comment (SC1) · 7 Mar 2017

This is a nice and interesting approach to fold back trajectories with land surface data for qualitative aerosol source apportionment. It strongly reminds me of some of our earlier work (van Pinxteren et al., 2010), where we derived a "residence time" parameter very similar to the "exposure" parameter described here and included it into a PCA as done here as well. We used HYSPLIT back trajectory ensembles, which might give somewhat coarser results than FLEXPART footprints, but nevertheless proved themselves valuable in a number of further qualitative source apportionment studies, including one on small-chain dicarboxylic acids (van Pinxteren et al., 2014). The authors might want to consider these papers and maybe reassess their statement on P10

[Figure]

L11-12 that such information cannot be derived from simple trajectories.

References

van Pinxteren, D., Brüggemann, E., Gnauk, T., Müller, K., Thiel, C., and Herrmann, H.: A GIS based approach to back trajectory analysis for the source apportionment of aerosol constituents and its first application, J. Atmos. Chem., 67, 1-28, doi: 10.1007/s10874-011-9199-9, 2010.

van Pinxteren, D., Neusüß, C., and Herrmann, H.: On the abundance and source contributions of dicarboxylic acids in size-resolved aerosol particles at continental sites in central Europe, Atmos. Chem. Phys., 14, 3913-3928, doi: 10.5194/acp-14-3913-2014, 2014.

---

## Referee Comment (RC2) · Anonymous Referee #2 · 16 Mar 2017

Reviewer comments: acp-2017-90, Exploring sources of biogenic secondary organic aerosol compounds using chemical analysis and the FLEXPART model

The authors quantify a number of organic compounds in PM2.5 samples collected on filters in rural Sweden, in particular several acids, di-acids, and organosulfates. By modeling (using FLEXPART) the landcover types that influenced each sample, the authors are able to use principle component analysis to connect landcover to composition and qualitatively determine the importance of different landcover types. This is, in general, a valuable and important goal, and a reasonable approach to doing so. The conclusions of this work add to the body of work demonstrating the importance of coniferous forests to European aerosol loadings, and the work is generally scientifically

rigorous with an honest assessment of capabilities and limitations. There are certain places in this work where the authors could dig deeper, discuss further implications, or further clarify and frame the proper interpretation of the data; these are discussed below and may make this analysis stronger, but these are generally only minor revisions.

General comments:

The crux of the scientific contribution of this work is the PCA, in particular the connection between landcover and composition, and the correlations between some of the straight-chain di-acids. However, the results of the PCA are not particularly surprising (biogenic products come from forests), while some substantial lingering questions that are not wholly addressed by the authors.:

1) The main conclusions revolve around PC1, which includes a large majority of the measured components, as well as most of the explained variability. As noted by the authors, the compounds include both monoterpene and isoprene products, which are known to be dominant emissions from different tree species –including further components may split this out (or not), did the authors consider trying other PC solutions or parameters?

2) For PC2 (and PC4 to a lesser extent), no compounds really correlate with these cover types – was there lower loadings in general, or was PM2.5 just comprised of compounds not measured in this work?

3) There are some biogenic compounds that don't really fall strongly in to any of the PCs (OS250, both NOS) – any thoughts about why that is or how those are different from all the others that co-vary? In the case of pinonic acid, for instance, previous work (Thompson et al., http://dx.doi.org/10.1080/02786826.2016.1254719) has shown it is mostly in the gas phase, so filter samples may be mostly gas-phase artifacts which may make it different than the other lower volatility products.

4) Was any attempt made to consider the age of the particle as it relates to landcover?

That is, a coniferous forest (or pixel) near the site will emit pinene, but not be turned in to pinic acid before the site, while a farther forest (or pixel) might. So all pixels are not created equal, and in many ways these factors are probably a driving force in variability. For instance, is it not possible for likely that the covariance of isoprene and monoterpene products that put them in the same PC is due to chemical processes, not differences in emission from landcover?

5) Throughout the work, the authors classify the di-acids into two groups, anthropogenic and fatty acid, but it's not clear why that is and given their structures why they wouldn't just get binned. Unless it is in the introduction and I missed it. This is especially confusing given that the authors go on to demonstrate that they co-vary, or at least if they are two different groups (adipic vs. others) it is not in the breakdown pre-supposed.

Specific comments:

P. 2 L. 2 – Why start the paper with a comment about PM10 when the rest of the paper is about PM2.5?

P. 2 L. 9 – "Four main categories. . ." This is an odd breakdown, they aren't really symmetrical categories. Isoprene isn't really a "category" it's just one compound, and there is no clear split between "other reactive" and "less reactive". So the categories are 'individual compound', 'class of compounds', 'compounds of a certain poorly defined property', 'compounds of a different certain poorly defined property'. It should be classes, or sources, or properties, or some unifying 'type' of categories. I can deal with isoprene being treated as a 'class' if there is generally other symmetry, but the switch of 'class' to 'property' is asymmetric.

P. 3 L. 1 – "BVOCs constitute more than 50% of all atmospheric VOCs. . ." – If I'm not mistaken, that is low, generally BVOCs are more like 90%.

P. 3 L. 9 – Use "Influence" instead of "enhancement". While the influence of sulfate is

generally enhancing, the presence of OSs only points to influence, they do not necessarily indicate that BSOA mass would have been lower in the absence of anthropogenic influence, just that it would have been different composition.

P. 4 L. 15 – Eluent A is 0.1% acetic acid in what? Water?

P.5. L. 14 – "A formal source apportionment would require a precise accounting of these factors, which is extremely complicated and is clearly out of the scope of this study." This is a subjective sentence that could be re-worded to more precisely state the reasons for not providing more discussion or detail into the impact of the factors discussed in the previous sentence. Even if "out of the scope," some discussion of these factors would greatly enhance the discussion and interpretation of these results, see General Comment 4. Change to something more like "A formal source apportionment that includes a precise accounting of these factors is out of the scope of this study which is focused on landcover types, but some discussion of these factors is included where relevant."

P. 5. L. 29 – put a separation in 100000, either a space as used elsewhere, or change all of them to commas (my preference, as an American. . .) or periods

P. 6 L. 1 – Typo in "ocean"

P. 6 L. 14 – Change to "PCA was. . ."

P.6 L. 20 – Should be "noting" instead of "noticing".

P.6 L. 21 – "most probably" is fairly informal, and "has" is the wrong tense. Can probably just combine this and the next sentence into one sentence.

P. 6 L. 26 – see General Comment 5, why are the di-acids split into different categories?

P. 7 L. 15 – Here, and in general, the discussion and analysis would be bolstered by testing for the effect of excluding these days with peaks. Does the correlation become more like that of the other acids? Does it fall into the same PC as the other acids? In

other words, is the big difference of this acid just these two peaks? And if so, is there any indication in the back trajectories or composition of what might be causing these peaks?

P. 7 L. 29 – The intuitive interpretation of these data is of course what the authors not should not be interpretation, that the land cover exposure is a measure of the contribution of the landcover to aerosol production. It would help for the authors to re-iterate what the proper interpretation is, since it is not wholly clear (note that in the methods section, the back trajectories are discussed in the subheading of "Source Apportionment"). This clarification may help shape the discussion somewhat

P. 8 L. 22 – Should be "Methods section"

P. 9 L. 6 – It would be interesting to see if adipic ended up in this factor if the 2 spikes were excluded (see comment below about PC4)

P. 9 L. 12 – Interpreting 0.21 has a meaning contribution is probably somewhat overinterpreting. Perhaps this is a place where exploring other correlations or factors may be worth discussion.

P. 9 L. 16 – Again, how much of this is due to those two spikes? To speculate for a moment, is it not possible that the landcover types associated with PC4 just happen to be co-located with some strong point source of adipic acid, so it is all due to an unidentified covariance?

P. 10 L. 9 – Remove "totally." Again, a comprehensive implementation may be out scope, but some more discussion of these factors is warranted in the PC analysis, particularly results that are difficult to explain by landcover.
* * *

---

## Referee Comment (RC3) · Anonymous Referee #3 · 27 Mar 2017

Review of Martinsson et al., "Exploring sources of biogenic secondary organic aerosol compounds using chemical analysis and the FLEXPART model"

Synopsis

In this work, the abundance and time variability of molecular compounds identified in filter samples of ambient air measured at the Vavihill site in Sweden is investigated. The authors then use a numerical Lagrangian particle dispersion model (FLEXPART) and statistical analysis (PCA) to identify source regions for the different compounds they have identified. They find that coniferous forests contribute strongly to monoterpene tracers found in organic aerosol samples.

[Figure]

General comments

The authors attempt to use backtrajectory calculations to qualitatively connect exposure of an air mass to land surface types to molecular markers found in organic aerosol samples, thereby investigating their formation formation processes. This is a nice and straightforward idea. The authors struggle, however, to convince the reviewer that their analysis and modeling has been conducted in a knowledgeable way. Their use of beta release software without justification, plus a number of other vaguenesses in the methods description are a warning sign that considerable caution should be taken before this manuscript can be published. Apart from questions regarding the methods used, the final results of the analysis don't seem to provide much new information. The fact that coniferous forest emits monoterpenes which then forms SOA has been shown numerous times. If there are other important findings, they are not apparent to the reviewer.

In summary I am tempted to recommend rejecting the manuscript due to the deficiencies in the methods section and the lack of scientifically new findings. It should be noted that in my review I cannot comment on aspects of the chemical analysis, as this is not my field of expertise. Hence I will recommend 'major revisions' here, as the chemical analysis might contain information that is novel for other readers.

Specific comments

* Use of beta software in analysis

The current stable release version of FLEXPART is 9.02, while the authors (claim to?) use version 10.0. Software in beta versions is considered unstable and for testing purposes only and can surely not be used in a scientific publication.

* HYSPLIT and FLEXPART together

It is unclear why simulations using the HYSPLIT model are 1) done at all and 2) presented as auxiliary analysis which is different from the FLEXPART analysis. Both HYS-

PLIT and FLEXPART solve the transport equations backwards in time. HYSPLIT as used here calculates single, deterministic trajectories, while FLEXPART calculates a large number (100000 in the present case) of trajectories, applying processes like turbulence and convection stochastically. FLEXPART by default delivers mass-weighted center trajectories and clusters (see documentation), which provide information equivalent to HYSPLIT. There is no additional information gained from the use of the HYSPLIT model, unless the authors start and compare the model results in detail. I recommend removing this completely, at most leaving a sentence stating that they evaluated HYSPLIT and it gave similar results.

* Uncertainties due to neglected sources and sinks during transport

On several occasions the authors caution that what they are doing is neither a full source inversion, nor a modeling effort considering (non-linear) effects of chemistry and other sources and sinks in the atmosphere. Statements like : "a formal source apportionment would require precise accounting of these factors, which is extremely complicated and is clearly out of scope of this study" (p 5, l 14-15) leave the reader wondering what this study is about, then, as more then half of the manuscript deals with exactly this kind of analysis on a simple level. This leaves the reader with the uneasy feeling that he/she cannot attribute significance to the findings. How large are those uncertainties? Where do they come from?

* "Surface vertical level" method

The "surface vertical level" definition as half the PBL height for each particle is not a standard FLEXPART output product that I am aware of (at least not in v 9.02). There is no documentation of this feature judging by a cursory look over the available publication (Stohl et al., 2005, ACP) and a quick source code survey. While surely useful, I don't see how the authors have achieved this without coding it themselves. This would have to be described accordingly, if this is what they did. Furthermore, the choice of 1/2 the PBLH is arbitrary, and the reasoning ("non-linear processes" again) is insufficient.

[Figure]

\* AERO-Tracer

Justify the use of the particle diameter used, as this has considerable effect on the lifetime of the particles and hence the exposure calculation. I suggest recalculating for large and small particles.

\* source apportionment

Given the description on page 6 (top paragraph), you are simply multiplying the response function output fields (units of s m3 kg-1) by the fractional land cover - did you correct for grid area and level thickness?

\* Principal component analysis (PCA) method

The method description of the PCA is insufficient. Citing a commercial, non-free software package is not an appropriate source of information for the reader.

Also: for a PCA to be meaningful, a number of preconditions have to be met, out of which I wonder if two are met: 1) sample size: 38 data points (filter samples) is quite small, can you show that the results are still reasonable? 2) outliers: did you remove them?

\* PCA results

The kind of PCA performed should be described in the methods section, see above.

---

## Author Comment (AC1) · 28 Jun 2017

**We would like to thank the reviewer for his/her helpful comments which improved the manuscript considerably. Answers to comments are written in blue. Changes in the manuscript are marked with red.**

**Review: Atmos. Chem. Phys. Discuss.; acp-2017-90 (Martinsson et al.)**

**General comments:**
This manuscript explores the connection between chemical speciation at the micro level and air mass surface exposure at the macro level. The study is based on a comprehensive chemical data set of organic species. An expected outcome of the study is that that the surface category "coniferous forest" had a clear impact on the mass concentration of the measured compounds, whereas the surface category "sea and ocean" only had a low explanatory power. As the authors state, the biogenic source and surface origin of the dicarboxylic acids, azelaic acid, suberic acid and pimelic acid, which are closely related, is not clear, and should be the focus of future studies.

**Specific comments:**
Page 2 – line 14: Another monoterpene to be considered is $\Delta_3$-carene (Räisänen et al., 2013).

We have added this information.

Page 2 – 25: The number of carbon atoms in a molecule should be indicated with a subscript: $C_3$-$C_6$, $C_7$-$C_9$.

This has been corrected.

Page 4 – line 23: A correction is needed here: "….. their MS/MS formation of $HSO_4^-$ ($m/z$ 97) ……….. (63 u)….". Note that according to the IUPAC guidelines for terms related to mass spectrometry "$m/z$" should be in italic font (Murphy et al. 2013). Furthermore, the neutral loss of $HNO_3$ corresponds to 63 "mass units", abbreviated as "u".

This has been corrected.

Page 16 – Table 1: The structure of MBTCA should be corrected as follows:

This has been corrected.

**Technical corrections:**
Page 2 – line 6 and many places elsewhere: a space is missing after ";" in the cited references.

An error in the citation software won't allow the authors to insert this space. We are aware of this and hope that it can be fixed through typesetting if the manuscript gets accepted.

Page 2 – line 16: …. gas-phase products …..

This has been corrected

Page 2 – line 17: ….. in the aerosol phase.

This has been corrected.

Page 3 – line 3: …. low-volatility …..

This has been corrected.

Page 3 – line 20: ….. a one-year study ….

The current study is not a one-year study. It stretches from June 2012-July 2012.

Page 4 – line 2: …. high-volume …..

This has been corrected.

Page 4 – lines 3, 5 and 12: a space is missing before "$^0C$".

This has been corrected.

Page 4 – line 9: …. 15 µL……

This has been corrected.

Page 4 – lines 16, 17 and 18: ….. min ….. ("minutes" is abbreviated as "min").

This has been corrected.

Page 4 – lines 9 and 11: ….. mL …….

This has been corrected.

Page 4 – line 19: The abbreviation "MS" stands for the technique "mass spectrometry" and not for theinstrument. Thus: "The ESI-q-TOF-MS instrument …." (see Murphy et al., 2013).

This has been corrected.

Page 5 – line 13: ….. of precipitation, …..

This has been corrected.

Page 7 – line 3: …. fatty acid-derived …..

This has been corrected.

Page 7 – line 12/13: …. fatty acid-derived …..

This has been corrected.

Page7 – line 13: …. have a different origin than isoprene- and monoterpene-generated acids, …..

This has been corrected.

Page 8 – line 3: …. "broad-leaved forest"…..

This has been corrected.

Page 8 – line 23: …… and monoterpenes …..

This has been corrected.

Page 8 – line 26: ….. monoterpene-derived …….. both monoterpene- and isoprene-derived …..

This has been corrected.

Page 8 – line 33: …. "broad-leaved forest"…..

This has been corrected.

Page 9 – line 12: …. "broad-leaved forest"…..

This has been corrected.

Page 9 – line 14: ….. ($C_7$-$C_9$) …. (see specific comment above).
This has been corrected.

Page 9 – line 19: …… is thought to …..

This has been corrected.

Pages 11 – 14: Titles of journal articles should not be capitalized.

This has been corrected.

Table 2 – legend: Measured *m/z*, ……. f) Surratt et al. (2008), ………… h) Surratt et al. (2010).

This has been corrected.

References:
K. K. Murray, R. K. Boyd, M. N. Eberlin, G. J. Langley, L. Li, Y. Naito. Definitions of terms relating to mass spectrometry (IUPAC Recommendations 2013). Pure Appl. Chem., 85, 1515-1609, 2013.

T. Räisänen, A. Ryyppö, S. Kellomäki. Effects of elevated $CO_2$ and temperature on monoterpene emission of Scots pine (Pinus sylvestris L.). Atmos. Environ. 2008, 42, 4160.

---

## Author Comment (AC2) · 28 Jun 2017

This is a nice and interesting approach to fold back trajectories with land surface data for qualitative aerosol source apportionment. It strongly reminds me of some of our earlier work (van Pinxteren et al., 2010), where we derived a "residence time" parameter very similar to the "exposure" parameter described here and included it into a PCA as done here as well. We used HYSPLIT back trajectory ensembles, which might give somewhat coarser results than FLEXPART footprints, but nevertheless proved themselves valuable in a number of further qualitative source apportionment studies, including one on small-chain dicarboxylic acids (van Pinxteren et al., 2014). The authors might want to consider these papers and maybe reassess their statement on P10 L11-12 that such information cannot be derived from simple trajectories.

We have removed the above-mentioned statement and included an acknowledgement to the study by van Pinxteren et al. 2010: *"van Pinxteren et al. (2010) demonstrated how air mass exposure to land cover affected the measured size-resolved organic carbon (OC), elemental carbon (EC) and inorganic compounds at a receptor site in Germany by using the HYSPLIT model."* This sentence is to be found in the introduction.

**References**

van Pinxteren, D., Brüggemann, E., Gnauk, T., Müller, K., Thiel, C., and Herrmann, H.: A GIS based approach to back trajectory analysis for the source apportionment of aerosol constituents and its first application, J. Atmos. Chem., 67, 1-28, doi: 10.1007/s10874-011-9199-9, 2010.

van Pinxteren, D., Neusüß, C., and Herrmann, H.: On the abundance and source contributions of dicarboxylic acids in size-resolved aerosol particles at continental sites in central Europe, Atmos. Chem. Phys., 14, 3913-3928, doi: 10.5194/acp-14-3913-2014, 2014.

---

## Author Comment (AC3) · 28 Jun 2017

Reviewer comments: acp-2017-90, Exploring sources of biogenic secondary organic aerosol compounds using chemical analysis and the FLEXPART model

The authors quantify a number of organic compounds in PM2.5 samples collected on filters in rural Sweden, in particular several acids, di-acids, and organosulfates. By modeling (using FLEXPART) the landcover types that influenced each sample, the authors are able to use principle component analysis to connect landcover to composition and qualitatively determine the importance of different landcover types. This is, in general, a valuable and important goal, and a reasonable approach to doing so.

The conclusions of this work add to the body of work demonstrating the importance of coniferous forests to European aerosol loadings, and the work is generally scientifically rigorous with an honest assessment of capabilities and limitations. There are certain places in this work where the authors could dig deeper, discuss further implications, or further clarify and frame the proper interpretation of the data; these are discussed below and may make this analysis stronger, but these are generally only minor revisions.

General comments:
The crux of the scientific contribution of this work is the PCA, in particular the connection between landcover and composition, and the correlations between some of the straight-chain di-acids. However, the results of the PCA are not particularly surprising (biogenic products come from forests), while some substantial lingering questions that are not wholly addressed by the authors:

1) The main conclusions revolve around PC1, which includes a large majority of the measured components, as well as most of the explained variability. As noted by the authors, the compounds include both monoterpene and isoprene products, which are known to be dominant emissions from different tree species –including further components may split this out (or not), did the authors consider trying other PC solutions or parameters?

We performed several PCAs by varying the number of rotated factors from 2 to 6. We then judged the interpretability of the PCAs by attempts to associate logical and physical explanations to the extracted factors. In this dataset, the subjective best interpretation was observed by using VARIMAX rotation for 4 extracted factors. We have added some information regarding this issue in the "2.4.3 Principal Component Analysis (PCA)" section.

2) For PC2 (and PC4 to a lesser extent), no compounds really correlate with these cover types – was there lower loadings in general, or was PM2.5 just comprised of compounds not measured in this work?

We, as authors, do not really understand this comment and question. The PCA loadings are displayed in Table 7, and they vary depending on which PC you study.

We found detectable concentrations of all presented compounds. Hence, the case might be that the strongly contributing land-cover types in PC2 and PC4 do not contribute to the observed PM-species.

3) There are some biogenic compounds that don't really fall strongly in to any of the PCs (OS250, both NOS) – any thoughts about why that is or how those are different from all the others that co-vary? In the case of pinonic acid, for instance, previous work (Thompson et al., http://dx.doi.org/10.1080/02786826.2016.1254719) has shown it is mostly in the gas phase, so filter samples may be mostly gas-phase artifacts which may make it different than the other lower volatility products.

Formation of NOS depends on availability of precursors, including NOx, which could affect their variation compared to other tracers. OS250 is a product of alpha/beta-pinene and it is not clear why the correlation to other OS is low.

We agree that a major fraction of pinonic acid is found in the gas phase, though the partitioning is expected to be somewhat shifted at the lower temperatures in Sweden compared to the study of Thompson et al. in Alabama during summer, favoring partitioning to the particle phase in this work. Previous work of Kristensen et al. (http://dx.doi.org/10.1016/j.atmosenv.2015.10.046) showed that as much as 80% of pinonic acid collected with a high volume sampler could be due to gas phase adsorption, but if gaseous and particulate products are transported in the same air masses this should only affect the variation to a minor degree.

4) Was any attempt made to consider the age of the particle as it relates to landcover?

That is, a coniferous forest (or pixel) near the site will emit pinene, but not be turned in to pinic acid before the site, while a farther forest (or pixel) might. So all pixels are not created equal, and in many ways these factors are probably a driving force in variability. For instance, is it not possible for likely that the covariance of isoprene and monoterpene products that put them in the same PC is due to chemical processes, not differences in emission from landcover?

This is a very good idea. It is a bit out of the scope of this study but we have added some sentences regarding this approach in the outlook.

Still, monoterpenes and isoprene are, as far as the authors are aware, derived from different types of forests (more specifically, plants). Hence, any similarity in emission patterns may portrait emissions from mixed forests while differences may indicate emissions from very specific land-cover types.

5) Throughout the work, the authors classify the di-acids into two groups, anthropogenic and fatty acid, but it's not clear why that is and given their structures why they wouldn't just get binned. Unless it is in the introduction and I missed it. This is especially confusing given that the authors go on to demonstrate that they co-vary, or at least if they are two different groups (adipic vs. others) it is not in the breakdown pre-supposed.

We have added some information on the precursor sources to these acids in the results and discussion section.

Specific comments:
P. 2 L. 2 – Why start the paper with a comment about PM10 when the rest of the paper is about PM2.5?

This has been changed.

P. 2 L. 9 – "Four main categories: : :" This is an odd breakdown, they aren't really symmetrical categories. Isoprene isn't really a "category" it's just one compound, and there is no clear split between "other reactive" and "less reactive". So the categories are 'individual compound', 'class of compounds', 'compounds of a certain poorly defined property', 'compounds of a different certain poorly defined property'. It should be classes, or sources, or properties, or some unifying 'type' of categories. I can deal with isoprene being treated as a 'class' if there is generally other symmetry, but the switch of 'class' to 'property' is asymmetric.

Very good comment. We have changed "categories" to "classes". We have also re-phrased the sentence and removed the last two "categories".

P. 3 L. 1 – "BVOCs constitute more than 50% of all atmospheric VOCs: : :" – If I'm not mistaken, that is low, generally BVOCs are more like 90%.

The numbers given are correct according to the cited references.

P. 3 L. 9 – Use "Influence" instead of "enhancement". While the influence of sulfate is generally enhancing, the presence of OSs only points to influence, they do not necessarily indicate that BSOA mass would have been lower in the absence of anthropogenic influence, just that it would have been different composition.

This has been corrected.

P. 4 L. 15 – Eluent A is 0.1% acetic acid in what? Water?

Yes, in water. This has been clarified.

P.5. L. 14 – "A formal source apportionment would require a precise accounting of these factors, which is extremely complicated and is clearly out of the scope of this study." This is a subjective sentence that could be re-worded to more precisely state the reasons for not providing more discussion or detail into the impact of the factors discussed in the previous sentence. Even if "out of the scope," some discussion of these factors would greatly enhance the discussion and interpretation of these results, see General Comment 4. Change to something more like "A formal source apportionment that includes a precise accounting of these factors is out of the scope of this study which is focused on landcover types, but some discussion of these factors is included where relevant."

Good comment. We have changed this in accordance to the reviewers comment.

P. 5. L. 29 – put a separation in 100000, either a space as used elsewhere, or change all of them to commas (my preference, as an American: : :) or periods

This has been corrected.

P. 6 L. 1 – Typo in "ocean"

This has been corrected.

P. 6 L. 14 – Change to "PCA was: : :"

This has been corrected.

P.6 L. 20 – Should be "noting" instead of "noticing".

This has been corrected.

P.6 L. 21 – "most probably" is fairly informal, and "has" is the wrong tense. Can probably just combine this and the next sentence into one sentence.

This has been corrected.

P. 6 L. 26 – see General Comment 5, why are the di-acids split into different categories?

We have added some information on the precursor sources to these acids in the results and discussion section.

P. 7 L. 15 – Here, and in general, the discussion and analysis would be bolstered by testing for the effect of excluding these days with peaks. Does the correlation become more like that of the other acids? Does it fall into the same PC as the other acids? In other words, is the big difference of this acid just these two peaks? And if so, is there any indication in the back trajectories or composition of what might be causing these peaks?

We have removed the concentration peaks in adipic acid and re-analyzed the data and the PCA. The outcome of this re-analysis is stated in section "3.3 Connection between surface type and measured species".

P. 7 L. 29 – The intuitive interpretation of these data is of course what the authors not should not be interpretation, that the land cover exposure is a measure of the contribution of the landcover to aerosol production. It would help for the authors to re-iterate what the proper interpretation is, since it is not wholly clear (note that in the methods section, the back trajectories are discussed in the subheading of "Source Apportionment"). This clarification may help shape the discussion somewhat.

We, as authors, are not sure what the intention of the reviewer's comment is here.

In the mentioned line we are discussing the exposure contribution of the "Other" category. We are stating that this category has significant impact on the exposure and a deeper discussion on the "Other" category follows a few rows down: "Further, the category "other" is also increased during this particular period….."

P. 8 L. 22 – Should be "Methods section"

This has been corrected.

P. 9 L. 6 – It would be interesting to see if adipic ended up in this factor if the 2 spikes were excluded (see comment below about PC4)

We have removed the concentration peaks in adipic acid and re-analyzed the data and the PCA. The outcome of this re-analysis is stated in section "3.3 Connection between surface type and measured species".

P. 9 L. 12 – Interpreting 0.21 has a meaning contribution is probably somewhat overinterpreting. Perhaps this is a place where exploring other correlations or factors may be worth discussion.

We agree with the reviewer that drawing conclusions from a loading of 0.21 is dangerous and may lead to false interpretations. Hence, we have removed the concluding sentence stating that broad-leaved forest may contribute to carboxylic acid production.

We have added more discussion regarding the carboxylic acids further down in the same section.

P. 9 L. 16 – Again, how much of this is due to those two spikes? To speculate for a moment, is it not possible that the landcover types associated with PC4 just happen to be co-located with some strong point source of adipic acid, so it is all due to an unidentified covariance?

We have removed the concentration peaks in adipic acid and re-analyzed the data and the PCA. The outcome of this re-analysis is stated in section "3.3 Connection between surface type and measured species".

P. 10 L. 9 – Remove "totally." Again, a comprehensive implementation may be out scope, but some more discussion of these factors is warranted in the PC analysis, particularly results that are difficult to explain by landcover.

We have removed "totally". Further, we have added more discussion in the 3.3 section. Especially, regarding the carboxylic acids and the effect of removing concentration peaks of adipic acids and the followed re-analysis.

---

## Author Comment (AC4) · 28 Jun 2017

Review of Martinsson et al., "Exploring sources of biogenic secondary organic aerosol compounds using chemical analysis and the FLEXPART model"

Synopsis
In this work, the abundance and time variability of molecular compounds identified in filter samples of ambient air measured at the Vavihill site in Sweden is investigated. The authors then use a numerical Lagrangian particle dispersion model (FLEXPART) and statistical analysis (PCA) to identify source regions for the different compounds they have identified. They find that coniferous forests contribute strongly to monoterpene tracers found in organic aerosol samples.

General comments
The authors attempt to use backtrajectory calculations to qualitatively connect exposure of an air mass to land surface types to molecular markers found in organic aerosol samples, thereby investigating their formation formation processes. This is a nice and straightforward idea. The authors struggle, however, to convince the reviewer that their analysis and modeling has been conducted in a knowledgeable way. Their use of beta release software without justification, plus a number of other vaguenesses in the methods description are a warning sign that considerable caution should be taken before this manuscript can be published. Apart from questions regarding the methods used, the final results of the analysis don't seem to provide much new information. The fact that coniferous forest emits monoterpenes which then forms SOA has been shown numerous times. If there are other important findings, they are not apparent to the reviewer.

In summary I am tempted to recommend rejecting the manuscript due to the deficiencies in the methods section and the lack of scientifically new findings. It should be noted that in my review I cannot comment on aspects of the chemical analysis, as this is not my field of expertise. Hence I will recommend 'major revisions' here, as the chemical analysis might contain information that is novel for other readers.

Specific comments
* Use of beta software in analysis
The current stable release version of FLEXPART is 9.02, while the authors (claim to?) use version 10.0. Software in beta versions is considered unstable and for testing purposes only and can surely not be used in a scientific publication.

The justification for using FLEXPART 10.0 over FLEXPART 9.02 is purely practical: we already had an installation of FLEXPART 10.0 (or based on FLEXPART 10.0: some of the

output subroutines were modified by us) running for another project, and it was simple to re-use it for computing these aerosol footprints.
We haven't encountered any instability, neither have we discovered any weirdness in the results that could point to a specific problem of aerosol simulations in FLEXPART 10.0. Therefore, unless there is such a known problem, we don't see the justification for re-computing the footprints with FLEXPART 9.02.

* HYSPLIT and FLEXPART together
It is unclear why simulations using the HYSPLIT model are 1) done at all and 2) presented as auxiliary analysis which is different from the FLEXPART analysis. Both HYSPLIT and FLEXPART solve the transport equations backwards in time. HYSPLIT as used here calculates single, deterministic trajectories, while FLEXPART calculates a large number (100000 in the present case) of trajectories, applying processes like turbulence and convection stochastically. FLEXPART by default delivers mass-weighted center trajectories and clusters (see documentation), which provide information equivalent to HYSPLIT. There is no additional information gained from the use of the HYSPLIT model, unless the authors start and compare the model results in detail. I recommend removing this completely, at most leaving a sentence stating that they evaluated HYSPLIT and it gave similar results.

We compute these HYSPLIT trajectories routinely, as auxiliary data of the measurements. They are, as the reviewer notices, very simplistic, and we didn't use them in the interpretation of the data. The FLEXPART-based analysis was done in a later stage, and is meant to be more thorough. We however agree that the interest of showing the HYSPLIT data is limited; therefore we removed them from Figure 2.

* Uncertainties due to neglected sources and sinks during transport
On several occasions the authors caution that what they are doing is neither a full source inversion, nor a modeling effort considering (non-linear) effects of chemistry and other sources and sinks in the atmosphere. Statements like: "a formal source apportionment would require precise accounting of these factors, which is extremely complicated and is clearly out of scope of this study" (p 5, l 14-15) leave the reader wondering what this study is about, then, as more then half of the manuscript deals with exactly this kind of analysis on a simple level. This leaves the reader with the uneasy feeling that he/she cannot attribute significance to the findings. How large are those uncertainties? Where do they come from?

Aerosols (some of them at least) contain reactive chemical species. The aerosol mass concentration of an air mass can change with factors such as the amount of solar radiation, the presence of reactive gas species in the air mass (OH, NOx, $O_3$, etc.), temperature and humidity, etc. Furthermore the aerosol mass concentration can itself influence the aforementioned parameters (i.e. reaction with chemical species will deplete these chemical species, it can change the albedo of the Earth in some wavelength ranges of the solar spectrum, which can in turn affect the temperature, etc.).

It is possible to attempt to reproduce these processes in a numerical model, and to use observations to evaluate the model results and possibly to provide an estimation of aerosols sources/sinks, in a top-down approach. This requires however 1) a more specialized model than FLEXPART, in particular one that can handle non-linear reaction between different transported species, 2) at least some prior knowledge on the aerosol production/destruction associated to each land surface type, and 3) a lot more observations. Two month of measurements of aerosol chemical composition at just one site is not nearly enough to provide constraints on aerosols sources and sinks, even locally.

The main objective of this paper is therefore not to produce an estimate of aerosol production, but to publish our measurements, in the hope that they will be useful to future studies. We attempted to interpret the data, within the limits of what is permitted by the size of the dataset: we chose a relatively simplistic modeling approach, and the scientific conclusions are limited, but it is unlikely that a more complex modeling approach would have led to more robust results. The bottleneck is the amount of data.

* "Surface vertical level" method

The "surface vertical level" definition as half the PBL height for each particle is not a standard FLEXPART output product that I am aware of (at least not in v 9.02). There is no documentation of this feature judging by a cursory look over the available publication (Stohl et al., 2005, ACP) and a quick source code survey. While surely useful, I don't see how the authors have achieved this without coding it themselves. This would have to be described accordingly, if this is what they did. Furthermore, the choice of 1/2 the PBLH is arbitrary, and the reasoning ("non-linear processes" again) is insufficient.

The feature is indeed not standard FLEXPART, we implemented it ourselves. It involves mainly two changes to the code:
- The PBL height at each particle position that is calculated in advance.f90 (in standard FLEXPART) is saved in an array.
- A new output module has been written, which accumulates in a 3D (lat, lon, time) array the residence time of particles between the surface and a user-defined threshold altitude, which can be either a fixed altitude or a fraction of the PBL. In the latter case, the height of these "virtual" surface grid-boxes varies from one place to another, and from one time-step to another, therefore we do not accumulate the residence time directly, but the residence times divided by the "virtual grid box" height and density, so that the resulting response function has a unit of $s.m^2/kg$.

These changes to FLEXPART were done for a different project, for which a manuscript is in preparation. The objective was to improve the representation of the diurnal variability in $CO_2$ and $CH_4$ simulations.

For these aerosol simulations, the impact is in fact negligible (the samples are taken over a 24 hours period, so the diurnal variability is smoothed), and that feature was just used as a default settings (there is no strong argument against or in favor of it). We recognize however that it should be better described and evaluated, and since this paper is not the good place for this, all the simulations in the revised manuscript use a more standard fixed surface level thickness of 400 m.

* AERO-Tracer

Justify the use of the particle diameter used, as this has considerable effect on the lifetime of the particles and hence the exposure calculation. I suggest recalculating for large and small particles.

FLEXPART distributes the particle diameters according to the mean diameter setting ("dquer", which we set to 250 nm), and to the "dsig" parameter (which controls the spread of the size distribution. We use a value of 12.5 for dsig, which means that the particles in a 250/12.5=20 nm to 12.5*250=3125 nm make 68% of the total particles mass.

The results are indeed dependent on the particle diameter. Smaller particles travel longer, and will therefore show higher sensitivity to remote land areas. We do not know accurately the size distribution of the particles we measured, but previous size distribution measurements at the measurement station during summer have shown a mean size distribution around a central value of 100 nm (Kristensson et al., 2008).

We experimented different mean particle diameters, i.e. we performed new simulations of 50 nm and 1 μm particles. However, these new simulations did not result in very different PCA results, at least not for the components which explained most of the variance in the data. PCA tables for particles of 50 nm and 1 μm are found in the supplement.

* Source apportionment
Given the description on page 6 (top paragraph), you are simply multiplying the response function output fields (units of s m3 kg-1) by the fractional land cover - did you correct for grid area and level thickness?
Yes (see previous point for the level thickness correction).

* Principal component analysis (PCA) method
The method description of the PCA is insufficient. Citing a commercial, non-free software package is not an appropriate source of information for the reader. Also: for a PCA to be meaningful, a number of preconditions have to be met, out of which I wonder if two are met: 1) sample size: 38 data points (filter samples) is quite small, can you show that the results are still reasonable? 2) outliers: did you remove them?

We have added more information about the PCA setup and performance in the methods-section.

1. We agree with the reviewer that it would have been desirable to have more samples than the 38 samples we have in this study. More samples would result in a more robust PCA and perhaps results that were easier to interpret. However, we believe that the current interpretation of the PCA output is sound, logical and satisfactory.

   We have not seen any strict recommendations regarding the number of samples vs. number of variables in PCA. By searching the literature we found that this ratio varies significantly between studies. In our study we had 38 samples and 32 variables (22 chemical species and 10 surface categories), that gives a sample-to-variable ratio of 1.19. van Pinxteren et al. (2010) used 29 samples and 60 variables (sample-to-variable ratio of 0.48). Viana et al. (2006) used 41 samples and 28 variables (sample-to-variable ratio of 1.46).

   Further, we have stressed the importance of larger datasets in future atmospheric PCA appliances. This is found in the discussion section.

2. Initially, we did not remove any potential outliers prior to the PCA. This is because we have no reason to distrust the data, even though it contains outliers. Outlier selection and removal is not trivial. Outliers can contain valuable information, in this particular case they can provide information regarding the sources of aerosols. The most obvious example of suspicious outliers in this dataset is the concentration of adipic acid which peaked during the 27[th] of June and 6[th] of July. Being aware of the potential disruption in the PCA caused by outliers, we removed the concentration peaks of adipic acid and re-analyzed the data. Hence, the manuscript now contains analysis with and without the adipic acid concentration peaks. A discussion on this matter is given in section 3.3 in the manuscript.

* PCA results
The kind of PCA performed should be described in the methods section, see above.

We have added more information about the PCA setup and performance in the methods-section.

References

Kristensson, A., Dal Maso, M., Swietlicki, E., Hussein, T., Zhou, J., Kerminen, V.-M., Kulmala, M. Characterization of new particle formation events at a background site in Southern Sweden: relation to air mass history. Tellus, 60B, 330-344, 2008.

van Pinxteren, D., Bruggemann, E., Gnauk, T., Muller, K., Thiel, C., and Herrmann, H.: A GIS based approach to back trajectory analysis for the source apportionment of aerosol constituents and its first application, J Atmos Chem, 67, 1-28, 10.1007/s10874-011-9199-9, 2010.

Viana, M., Querol, X., Alastuey, A., Gil, J. I., and Menendez, M.: Identification of PM sources by principal component analysis (PCA) coupled with wind direction data, Chemosphere, 65, 2411-2418, 10.1016/j.chemosphere.2006.04.060, 2006.